# Machine Learning Application for Rupture Risk Assessment in Small-Sized Intracranial Aneurysm

**DOI:** 10.3390/jcm8050683

**Published:** 2019-05-15

**Authors:** Heung Cheol Kim, Jong Kook Rhim, Jun Hyong Ahn, Jeong Jin Park, Jong Un Moon, Eun Pyo Hong, Mi Ran Kim, Seung Gyu Kim, Seong Hwan Lee, Jae Hoon Jeong, Sung Won Choi, Jin Pyeong Jeon

**Affiliations:** 1Department of Radiology, Hallym University College of Medicine, Chuncheon 24252, Korea; khc@hallym.or.kr; 2Department of Neurosurgery, Jeju National University College of Medicine, Jeju 63241, Korea; pedineur@daum.net; 3Department of Neurosurgery, Hallym University College of Medicine, Chuncheon 24252, Korea; sparkahn@naver.com; 4Department of Neurology, Konkuk University Medical Center, Seoul 05030, Korea; medicalstory@gmail.com; 5Department of Neurosurgery, National Medical Center, Seoul 04564, Korea; nsdocmoon@gmail.com; 6Molecular Neurogenetics Unit, Center for Genomic Medicine, Massachusetts General Hospital, Boston, MA 02114, USA; ephong0305@gmail.com; 7Buzzpole Inc., Seoul 04781, Korea; ran@buzzpole.com (M.R.K.); seung@buzzpole.com (S.G.K.); usim@buzzpole.com (S.H.L.); jman@buzzpole.com (J.H.J.); wociety@buzzpole.com (S.W.C.); 8Institute of New Frontier Stroke Research, Hallym University College of Medicine, Chuncheon 24252, Korea; 9Genetic and Research Inc., Chuncheon 24252, Korea

**Keywords:** intracranial aneurysm, convolutional neural network, subarachnoid hemorrhage

## Abstract

The assessment of rupture probability is crucial to identifying at risk intracranial aneurysms (IA) in patients harboring multiple aneurysms. We aimed to develop a computer-assisted detection system for small-sized aneurysm ruptures using a convolutional neural network (CNN) based on images of three-dimensional digital subtraction angiography. A retrospective data set, including 368 patients, was used as a training cohort for the CNN using the TensorFlow platform. Aneurysm images in six directions were obtained from each patient and the region-of-interest in each image was extracted. The resulting CNN was prospectively tested in 272 patients and the sensitivity, specificity, overall accuracy, and receiver operating characteristics (ROC) were compared to a human evaluator. Our system showed a sensitivity of 78.76% (95% CI: 72.30%–84.30%), a specificity of 72.15% (95% CI: 60.93%–81.65%), and an overall diagnostic accuracy of 76.84% (95% CI: 71.36%–81.72%) in aneurysm rupture predictions. The area under the ROC (AUROC) in the CNN was 0.755 (95% CI: 0.699%–0.805%), better than that obtained from a human evaluator (AUROC: 0.537; *p* < 0.001). The CNN-based prediction system was feasible to assess rupture risk in small-sized aneurysms with diagnostic accuracy superior to human evaluators. Additional studies based on a large data set are necessary to enhance diagnostic accuracy and to facilitate clinical application.

## 1. Introduction

The prevalence of intracranial aneurysms (IA) has been reported to be as high as 3.2 percent [1,2]. However, the rate of subarachnoid hemorrhage (SAH) due to aneurysm rupture is less than 2%, which means that only a small number of patients experience rupture events [3]. Nevertheless, if a rupture occurs, mortality rates up to 57% within the first six months have been reported [4,5]. Accordingly, the identification of unruptured intracranial aneurysms (UIA) and the risk assessment of future rupture events are important in treatment planning. In clinical circumstances, aneurysm location and size are the main factors influencing UIA treatment. The annual rupture rate of aneurysms in the internal carotid artery (ICA), the posterior communicating (P-com) artery, and the middle cerebral artery (MCA) is 1.6%. In contrast, UIA patients with aneurysms in the anterior cerebral artery (ACA), including the main anterior communicating (A-com) artery and the posterior cerebral artery, had higher annual rupture rates of up to 1.9% and 4%, respectively [6,7]. Size is an important determinate for rupture risk. Aneurysm sizes of 7 mm and greater have been identified as risk factors increasing lifelong rupture risk [8]. Due to technical advances and more detailed radiologic findings, such as irregular aneurysm walls, multi-lobulation, and hemodynamic patterns focusing on aneurysm wall shear stress (WSS) have gained attention [9]. However, many uncertainties remain regarding rupture risk, particularly in small saccular aneurysms less than 7 mm in size.

Recently, a deep learning algorithm using a convolutional neural network (CNN) has been increasingly applied to a broad variety of medical imaging modalities. Aubreville et al. [10] reported that a CNN showed a mean accuracy of 88.3% (sensitivity 86.6% and specificity 90%) for the automatic classification of cancerous tissue in confocal laser endomicroscopic image sequences of the oral cavity. Esteva et al. [11] also demonstrated that a CNN using only pixels and disease labels as input variables provided detection of skin cancer comparable to dermatologists. A CNN has also been used for automated aneurysm detection [12] and measurement in patients with IA [13]. Nakao et al. [12] reported that aneurysm detection rate by the CNN based on a maximum intensity projection (MIP) algorithm of magnetic resonance angiography (MRA) was 94.2%, with 2.9 false positives per case. In addition, the CNN measured aneurysm areas correctly more often than a radiologist [13].

Beyond these findings, another important issue for CNN application is to distinguish between aneurysms that have ruptured, and those that have not. Rupture status can be easily diagnosed by hemorrhage patterns on the computed tomography (CT), however, multiple aneurysms have been reported in up to 33.5% of patients [2,14,15]. In addition, in patients with multiple aneurysms, 31% harbored three or more (120 of 387) aneurysms. Since inaccurate identification of a ruptured aneurysm can lead to re-bleeding events [16], accurate assessment of a rupture is essential. Therefore, we have applied a CNN to images of three-dimensional (3D) digital subtraction angiography (DSA) to evaluate the rupture status in patients with small-sized aneurysms of the anterior circulation.

## 2. Materials and Methods

### 2.1. Study Population

This study included retrospective data collection and a prospective study in patients harboring IA at multiple centers between January 2012 and December 2018. Patients with a clinical diagnosis and neuroimaging findings of aneurysm morphology were recruited. The inclusion criteria in this study were: (1) adult patients over 18 years old, (2) a diagnosis of saccular aneurysms, (3) an IA located on the anterior circulation, and (4) small aneurysms with less than 7 mm maximal diameters. The exclusion criteria were: (1) a diagnosis of non-saccular aneurysms, such as fusiform or dissecting aneurysms; (2) aneurysms due to trauma or infection; (3) an IA on the posterior circulation; and (4) patients with multiple aneurysms. A retrospective data set composed of IA patients who underwent DSA between January 2012 and December 2016 was used as a training cohort for the CNN and the AlexNet architecture. The resulting CNN was then prospectively tested in IA patients who underwent 3D-DSA (3D-digital subtraction angiography) from January 2017 to December 2018 [17]. The aims of this investigation were to develop an automatic classification system to distinguish ruptured from unruptured aneurysms using 3D-DSA images, and to demonstrate the feasibility of the system using a training and validation test set. The clinical and radiologic data were reviewed by two readers (KHC and JPJ). This study was approved by the Institutional Review Board (No. 2018-11-008).

### 2.2. Image Acquisition and Data Processing

DSA was conducted with the Allura Xper FD 20/20 (Philips Medical System, Best, The Netherlands) biplane system and the Axiom Artis zee (Siemens Healthcare, Erlangen, Germany) in conjunction with standard injection protocols [18,19]. For 3D-DSA, a total of 20 mL of Iodixanol 320 (Visipaque, GE Healthcare, Chicago, IL, USA) was administered at 4 mL/s for 12 seconds and 121 projections were obtained in 240° within 4 seconds to construct 3D images. There was a 1.5 second time delay for contrast filling in the Allura Xper FD 20/20 system. For the Axiom Artis zee, a total of 18 mL of Pamiray 250 contrast agent (Iopamidol, Dong-kook Pharmaceutical, Seoul) was administered with a tube rotation time of 4.5 seconds. The images were reconstructed using a 256 × 256 matrix. Post-processing of 3D-DSA was done in an independent workstation equipped with InSpace 3D software [18,19,20]. Aneurysm images in six directions (anteroposterior, posteroanterior, both laterals, superoinferior and inferosuperior) were obtained. Regions-of-interest (ROIs) selected by the neuroradiologist which included the aneurysm were then applied to each image, as seen in Figure 1.

### 2.3. Neural Network Architecture for Aneurysm Images

A CNN with an Alexnet_v2 architecture was used. For the third through the fifth convolution layers, 3 × 3 sized filters were used. For the second layer and the first layer, 5 × 5 and 11 × 11 sized filters were used, respectively. The system received three channel 224 × 224 sized input images extracted with a diameter of 65 pixels, consisting of three max pooling layers, two drop-out layers, three fully connected layers, and five convolution layers as seen in Figure 2. The model was implemented on a TensorFlow API (Google Inc, Mountain View, CA, USA). Data sets were completely balanced (on categories) and resized to 224 × 224 as train data sets. In addition, data augmentation was performed. We flipped each piece of data horizontally and vertically. Accordingly, we were able to get four times more trained images in total compared to the original trained images. All parameters were trained from scratch using the Adam optimization method with a batch size of 20 on the Geforce RTX 2080 GPU (NVIDIA, Santa Clara, CA, USA). The learning rate was set to be 5 × 10^−7^ and the drop-out rate was 0.5. The Adam optimizer was set with the default parameters. Ruptured aneurysms were defined as those with an expected rupture risk of ≥50% among the aneurysm images after discussion.

### 2.4. Statistical Analyses

Descriptive results were presented as the number of subjects (percentage) for both discrete and categorical variables, and as the mean and standard deviation (SD). A two-by-two table was generated to assess the sensitivity, specificity, and overall accuracy of the results [18]. Prospective data of the test cohort obtained by the CNN or by the human evaluator were analyzed using receiver operating characteristics (ROC). Any *p*-values less than 0.05 were considered statistically significant. The analysis was conducted using SPSS version 19 (IBM, Armonk, NY, USA) and MedCalc (www.Medcalc.org).

## 3. Results

### 3.1. Clinical Characteristics of the Enrolled Patients

A total of 640 patients with IA were included. There were 368 patients in the retrospective training data set and 272 patients in the prospective test group as seen in Table 1. There are no differences in the number of females, incidence of hypertension, diabetes mellitus, hyperlipidemia, or smoking between the two groups as seen in Table 1. The number of SAHs present was 244 (66.3%) in the training cohort and 191 (70.2%) in the test cohort. The number of aneurysm locations did not reach statistical significance between the two groups (*p* = 0.265).

### 3.2. Accuracy of CNN for the Diagnosis of Ruptured Aneurysm

In the test cohort, the CNN demonstrated a sensitivity of 78.76% (95% CI: 72.30%–84.30%), a specificity of 72.15% (95% CI: 60.93%–81.65%), and an accuracy of 76.84% (95% CI: 71.36%–81.72%) to predict ruptured aneurysms, as seen in Table 2. The AUROC for the CNN was 0.755 (95% CI: 0.699%–0.805%) and was significantly better (*p* < 0.001) than that of the human evaluator (AUROC = 0.537; 95% CI: 0.476%–0.598%), as seen in Figure 3. We further evaluated accuracy according to aneurysm locations. For IA patients with ACA aneurysms, the CNN demonstrated a sensitivity of 86.36% (95% CI: 75.69%–93.57%), a specificity of 36.67% (95% CI: 19.93%–56.14%), and an accuracy of 70.83 percent (95% CI: 60.67%–79.67%). For MCA aneurysms, the CNN showed a sensitivity of 85.42% (95% CI: 72.24%–93.93%) and a specificity of 79.41 percent (95% CI: 62.10%–91.30%). Regarding ICA aneurysms, the sensitivity and specificity were 90.00% (95% CI: 79.49%–96.24%) and 55.88% (95% CI: 37.89%–72.81%), respectively.

## 4. Discussion

To the best of our knowledge, this is the first study that applied a CNN to assess rupture risk in small-sized aneurysms. Overall, the CNN had a sensitivity of 78.76% (95% CI: 72.30%–84.30%) and a specificity of 72.15% (95% CI: 60.93%–81.65%) for predicting ruptures in small-sized aneurysms. In addition, the CNN showed higher AUROC than experienced human evaluators (0.755 and 0.537, respectively).

Anatomical and hemodynamic characteristics have been widely studied to identify morphological parameters associated with IA rupture risk. Aneurysms are classified into several groups by their size: small (<7 mm), medium (7–12 mm), large (13–24 mm), and giant (≤25 mm) [21]. The first international study of unruptured intracranial aneurysms (ISUIA) reported that the rupture risk in patients with IA with a size of less than 10 mm, was less than 0.05% per year [22]. The second study of ISUIA also reported that IA patients who had aneurysms with a size of less than 7 mm without previous SAH histories had a 0% 5-year cumulative rupture rate in anterior circulation aneurysms of the ACA, MCA, and ICA, as well as a 2.5% 5-year cumulative rupture rate in posterior circulation and P-com aneurysms [23,24]. In contrast, patients with medium-sized UIAs in anterior circulation aneurysms, with the exception of cavernous aneurysms, had rupture rates of 1.5% and of 3.4% in posterior circulation and P-com aneurysms, respectively. Accordingly, it is reasonable to observe small aneurysms using repeat radiologic tests in patients without risk factors, such as prior SAH history, the presence of P-com or daughter sacs [25]. However, very small aneurysms (<5 mm) can rupture. Dolati et al. [24] reported that the rupture rate of aneurysms <5 mm was 37% among SAH patients. In hemodynamic studies, lower WSS and high oscillatory shear index (OSI) were associated with aneurysm ruptures (AUC = 0.85, 95% CI: 0.78–0.93) [26]. The authors suggested that increased permeability of the endothelium caused by enhanced endothelial adhesion molecules in lower WSS and high OSI could lead to aneurysm formation and degradation of the wall [27,28]. However, Hassan et al. [29] reported that higher WSS was related to IA growth and rupture in wide-necked aneurysms. These conflicting results suggested that higher WSS may initiate aneurysm formation and that lower WSS may aggravate aneurysm growth and subsequent rupture [30].

In recent years, CNNs have been used for various neuroradiology imaging findings, focusing on the development of efficient automatic diagnoses of solid tumors and vascular diseases. Laukamp et al. [31] reported that the multiparametric deep learning model (DLM) accurately diagnosed meningiomas in 55 out of 56 cases which correlated strongly (0.81) with manual segmentation measurements of tumor volumes. Stember et al. [13] demonstrated the highly accurate detection of IAs (85/86, 98.8%) using a CNN in conjunction with an MRA. In their study, 2D-MIP images were used for validation. Accordingly, some aneurysms located on or near tortuous parent arteries led to focal localization [13]. Nakao et al. [12] also reported the high accuracy of an automatic CNN system (94.2%, 98/104) to detect IAs. They reconstructed MRA images with a 0.469 mm isotropic voxel size and extracted each voxel inside or outside the arterial region from a volume of interest (VOI) along nine axes of MIP images [12]. In our study, we focused on predicting aneurysm rupture, not on diagnosis of the aneurysm itself. Although MRAs showed a high pooled sensitivity of 95% (95% CI: 89%–98%) in IA detection [32], DSA is widely accepted as a key radiologic examination used to identify delicate aneurysm morphologies and to set treatment plans. In addition, 3D-DSA provides clearer information on aneurysm characteristics and their relationship to nearby arteries than standard 2D or rotational DSA [33]. We collected DSA images in six directions in each patient and extracted images from the selected ROI. ROI images were then trained with the CNN classifier using an Alexnet_v2 architecture as seen in Figure 2. The output layer was a single unit and each image was assigned 0 or 1, according to the probability of being a ruptured aneurysm [12]. In the test cohort, the decision was made by the rupture rate in six images, whereby ≥50% indicated a positive finding. Compared to a neuro-interventionist, the CNN classifier showed better diagnostic accuracy to predict aneurysm rupture (AUROC difference = 0.218; *p* < 0.001).

Our system can be helpful to identify culprit aneurysms which can lead to SAH, particularly in patients harboring multiple aneurysms. The incidence of multiple aneurysms has been reported to be over 30% [2,15]. Nehls et al. [15] reported the common locations of multiple aneurysms in the order of P-com (21.5%), A-com (12%), and the ophthalmic artery (11%). Although MCA was the lowest site for rupture events in their study [15], identification of the culprit aneurysm was based on the distribution of the hemorrhage and the aneurysm’s morphologies. Nevertheless, the results could be arbitrary and largely based on the neurosurgeon’s experience. In our study, the AUROC of an experienced neuro-interventionist was 0.537 (95% CI: 0.476%–0.598%), indicating poor diagnostic accuracy for the detection of ruptured aneurysms. Accordingly, our CNN classifier enhanced the diagnostic accuracy for culprit aneurysms.

In this study, we performed additional trials using different combinations. First, we switched the CNN network to a Residual neural network (ResNet) 50 under the same conditions. We then used a pretrained ImageNet ResNet50 model without data augmentation. Second, we applied a histogram of oriented gradients (HOG) method on our training images without data augmentation trained with Alexnet_v2. Overall diagnostic accuracy was 60.53% in ResNet-50 and 65.17% in HOG + Alexnet, showing lower than Alexnet_v2 with data augmentation. 

There were some limitations to this study. First, we only included anterior circulation aneurysms. Anterior circulation aneurysms are more prevalent than posterior circulation aneurysms, at an approximate ratio of 7:3. In addition, the morphometric factors between anterior and posterior circulation aneurysms are different [34]. Tykocki et al. [34] reported that the aspect ratio and parent artery size were significantly different between patients with anterior and posterior circulation aneurysms. Accordingly, further studies to assess aneurysm rupture are warranted for aneurysms of the posterior circulation. Second, differences in angiographic machines were not considered in the interpretation of our results. Moreover, the criteria for choosing radiologic tests for IA are somewhat different. Third, we applied user-selected ROI around the aneurysm, therefore, the system was not a fully automatic classifier. A previous automatic detection system [12] also determined VOI empirically for image extraction. The CNN with an Alexnet_v2 network used in this study trains faster compared to earlier network models using sigmoid function. However, compared to other network architectures such as Visual geometry group (VGG)-16, the Alexnet network retains more information of unrelated background in the final convolutional layer [35]. Accordingly, we applied user-selected ROI around the aneurysm. Our results showed a sensitivity of 78.76%, a specificity of 72.15%, and an accuracy of 76.84% in predicting ruptured aneurysm. Nevertheless, further studies are needed to compare deep neural networks in large datasets and automatic ROI detection of aneurysms.

## 5. Conclusions

We proposed a computer-assisted detection system to assess aneurysm rupture probability using a CNN based on 3D-DSA images. Our system was feasible to detect the rupture of small-sized aneurysms and demonstrated superior diagnostic accuracy compared to human evaluators. Future studies, based on a large data-set including posterior circulation aneurysms, are required to increase the diagnostic accuracy and to facilitate its clinical application.

## Figures and Tables

**Figure 1 jcm-08-00683-f001:**
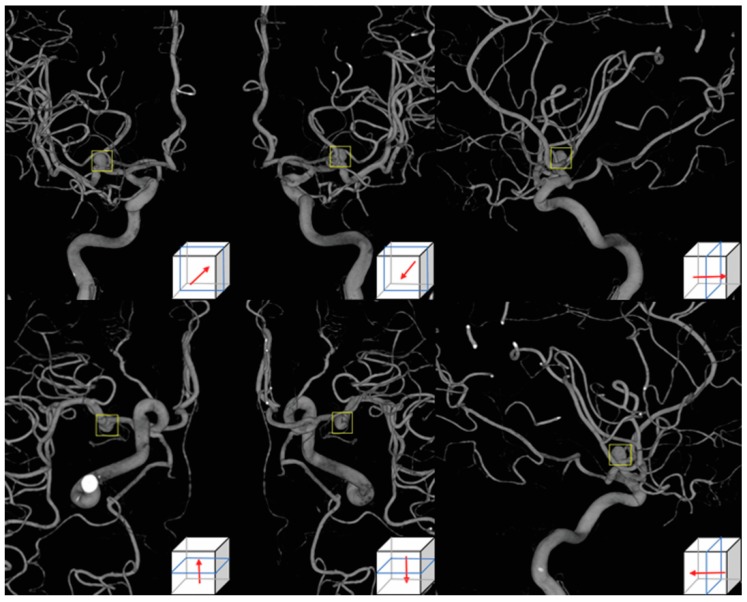
Aneurysm images using 3D-digital subtraction angiography in six directions, anteroposterior, posteroanterior, left lateral, superoinferior, inferosuperior, and the right lateral side. A user-selected region-of-interest shown as a yellow square was applied to each image.

**Figure 2 jcm-08-00683-f002:**
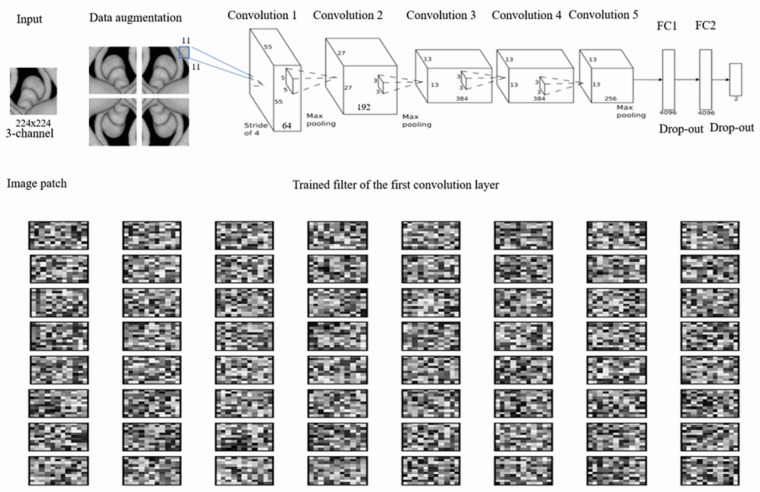
The proposed convolutional neural network consisting of convolutional layers, max-pooling layers, fully-connected layers, drop-out layers, and final layers of classification. The number of filters, the connections in each layer, and the first layer of the learned convolutional kernel are described above. FC, fully-connected.

**Figure 3 jcm-08-00683-f003:**
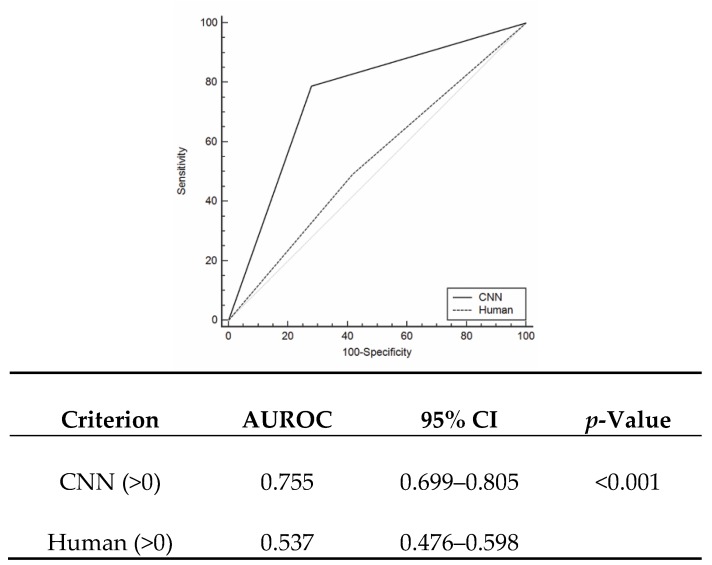
Receiver operating characteristics (ROC) curve according to detections by the convolutional neural network (CNN) and a human evaluator. The difference between areas is 0.163 (*p* < 0.001). Area under the ROC (AUROC) curve.

**Table 1 jcm-08-00683-t001:** Comparison of clinical and radiologic characteristics between the training and test cohorts.

	Training Cohort	Test Cohort	
Variables	(*n* = 368)	(*n* = 272)	*p*-Value
Clinical findings			
Female	175 (47.6%)	145 (53.3%)	0.150
Age, years	57.8 ± 14.4	55.8 ± 16.3	0.101
Hypertension	89 (24.2%)	72 (26.5%)	0.510
Diabetes mellitus	37 (10.1%)	28 (10.3%)	0.921
Hyperlipidemia	38 (10.3%)	21 (7.7%)	0.260
Smoking	44 (12.0%)	27 (9.9%)	0.419
Radiologic findings			
Lesion side, left	172 (46.7%)	127 (46.7%)	0.990
SAH presentation	244 (66.3%)	193 (71.0%)	0.294
Size (mm)	5.3 ± 1.0	5.2 ± 1.1	0.231
Territory location			
Anterior cerebral artery	123 (33.4%)	96 (35.3%)	0.265
Middle cerebral artery	133 (36.1%)	82 (30.1%)	
Internal cerebral artery	112 (30.5%)	94 (34.6%)	

Subarachnoid hemorrhage (SAH). Data are shown as the numbers of subjects (percentage) for discrete and categorical variables and mean ± standard deviation.

**Table 2 jcm-08-00683-t002:** Accuracy of the CNN to detect ruptured or unruptured intracranial aneurysms in the test cohort ^a^.

	Ruptured	Unruptured	Total
Ruptured	152	22	176
Unruptured	41	57	96
Total	193	79	272

^a^ The numbers given are patients.

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
