# Peer review of "Machine Learning Application for Rupture Risk Assessment in Small-Sized Intracranial Aneurysm"

_jcm, 2019, doi:10.3390/jcm8050683_

Reviewer 1 Report

In this manuscript, the authors developed a CAD system for small-sized aneurysm ruptures using a convolutional neural network (CNN) based on images of 3D digital subtraction angiography. They achieved an AUC of 0.706. I have the following comments:

-The authors proposed a simple CNN model in this manuscript that yields an AUC less than 0.80. To improve the results, the authors can use a deeper network, an ensemble of CNN models, fine-tune pre-trained models,  or residual layers. They also can apply data augmentation in the training phase.

-It is not clear how the authors extracted the ROIs (line 105), who did extract the ROIs? ROI size? and how many ROIs they extracted? In the caption of Fig.1, the authors stated ''A user-selected region-of-interest, shown as a yellow circle'' but I see squares, not circles.  In addition, the sentence of line 105 is fuzzy ''hen, user-selected regions-of-interest''.

-The description of the data is not clear. The authors should all details of the dataset and show more examples in the manuscript.

-The authors should compare their method with other related methods. In addition, the should compare the CNN model with texture analysis methods (LBP, Gabor, HOG, etc.).

-The quality of figures 2 and 3 should be improved.

-Please add a list of abbreviations and symbols at the end of the manuscript.

-Please improve the conclusion of the manuscript by adding the advantages of the method, the disadvantages of the other methods, best results and future work.                              

Author Response

Comment 1: The authors proposed a simple CNN model in this manuscript that yields an AUC less than 0.80. To improve the results, the authors can use a deeper network, an ensemble of CNN models, fine-tune pre-trained models, or residual layers. They also can apply data augmentation in the training phase.

Answer: Thank you for your careful review. Per your recommendation, we have further evaluated outcomes using various methods such as Residual neural network (ResNet)-50 model pretrained on the ImageNet dataset, Histogram of oriented gradients (HOG)+Alexnet and Alexnet V2 with data augmentation. Overall diagnostic accuracies were 60.53% in ResNet-50, 65.17% in HOG+Alexnet, and 76.84% in Alexnet V2 with data augmentation. Accordingly, we have revised results of studies performed using Alexnet with data augmentation. The difference in area under the ROC curve between CNN and a human evaluator was 0.218, which was higher than the previous difference of 0.163 (Figure shown below).                                             

Criterion

AUROC  

95%   CI

p-value

CNN   (>0)

0.755

0.699-0.805

<0.001< span="">

Human   (>0)

0.537

0.476-0.598

Comment 2: It is not clear how the authors extracted the ROIs (line 105), who did extract the ROIs? ROI size? and how many ROIs they extracted? In the caption of Fig.1, the authors stated ''A user-selected region-of-interest, shown as a yellow circle'' but I see squares, not circles.  In addition, the sentence of line 105 is fuzzy ''hen, user-selected regions-of-interest''.

Answer: Regions-of-interest (ROI) was selected by the neuroradiologist with 65 pixels in size for each image. For one patient, six ROIs for six-directional images were extracted. In Figure 1, we have revised the description as follows:

A user-selected region-of-interest shown as a yellow square was applied to each image.

Comment 3: The description of the data is not clear. The authors should all details of the dataset and show more examples in the manuscript.

Answer:  Per your recommendation, we have revised the methods as follows:

Post-processing of 3D-DSA was done in an independent workstation equipped with InSpace 3D software. Aneurysm images in six directions (anteroposterior, posteroanterior, both laterals, superoinferior and inferosuperior) were obtained. Regions-of-interest (ROIs) selected by the neuroradiologist which included the aneurysm were then applied to each image (page 3, line 101-106).

A CNN with an Alexnet_v2 architecture was used. For the third through the fifth convolution layers, 3x3 sized filters were used. For the second layer and the first layer, 5x5 and 11x11 sized filters were used, respectively. The system received three channel 224x224 sized input images extracted with a diameter of 65 pixels consisting of three max pooling layers, two drop-out layers, three fully connected layers, and five convolution layers. The model was implemented on Tensorflow API (Google Inc, Mountain View, CA, USA). Data sets were completely balanced (on categories) and resized to 224x224 as train data sets. In addition, data augmentation was performed. We flipped each piece of data horizontally and vertically. Accordingly, we could get four times more trained images in total compared to original trained images. All parameters were trained from scratch using Adam optimization method with a batch size of 20 on Geforce RTX 2080 GPU. The learning rate was set to be 5x10-7 and the drop-out rate was 0.5. As for Adam optimizer, it was set with default parameter values (page 3, line 113-124). The proposed convolutional neural network is described below (Figure 2 in the manuscript). 

Figure 2. The proposed convolutional neural network consisting of convolutional layers, max-pooling layers, fully-connected layers, drop-out layers, and final layers of classification

Comment 4: The authors should compare their method with other related methods. In addition, they should compare the CNN model with texture analysis methods (LBP, Gabor, HOG, etc.).

Answer: In this study, we performed additional trials using different combinations. First, we switched the CNN network to Resnet50 under the same condition. We then used a pretrained ImageNet Resnet50 model without data augmentation. Second, we applied HOG method on our training images without data augmentation trained with Alexnet. Overall diagnostic accuracy was 60.53% in ResNet-50 and 65.17% in HOG+Alexnet, showing lower than Alexnet_v2 with data augmentation. We have discussed these results in the discussion section (Page 7, line 237-242).

Comment 5: The quality of figures 2 and 3 should be improved.

Answer: Per your recommendation, we have improved qualities of these figures in the revised manuscript. 

Comment 6: Please add a list of abbreviations and symbols at the end of the manuscript.

Answer: Per your recommendation, we have added a list of abbreviations and acronyms at the end of the manuscript.

Abbreviations and Acronyms

ACA: Anterior cerebral artery; A-com: Anterior communicating artery; AUROC: Area under the receiver operating characteristics; CI: Confidence interval; CNN: Convolutional neural network; CT: Computed tomography; DLM: Deep learning model; DSA: Digital subtraction angiography; HOG: Histogram of oriented gradients; IA: Intracranial aneurysm; ICA: Internal carotid artery; ISUIA: International study of unruptured intracranial aneurysms; MCA: Middle cerebral artery; MIP: Maximum intensity projection; MRA: Magnetic resonance angiography; OSI: Oscillatory shear index; P-com: Posterior communicating artery; ROC: Receiver operating characteristics; ROI: Regions-of-interest; ResNet: Residual neural network;

SAH: Subarachnoid hemorrhage; SD: Standard Deviation; 3D: Three-dimension; UIA: Unruptured intracranial aneurysm; VGG: Visual geometry group; VOI: Volume of interest; WSS: Wall shear stress

Comment 7: Please improve the conclusion of the manuscript by adding the advantages of the method, the disadvantages of the other methods, best results and future work.

Answer: In this study, we used a CNN with an Alexnet_v2 network architecture. The algorithm is composed of five convolution layers and three fully connected layers. It trains faster compared to earlier network models using sigmoid function. However, compared to other network architectures such as Visual geometry group (VGG)-16, Alexnet network retains more information of unrelated background in the final convolutional layer (Reference 1 below). Accordingly, we applied user-selected ROI around the aneurysm. Our results showed a sensitivity of 78.76%, a specificity of 72.15%, and an accuracy of 76.84% in predicting ruptured aneurysm. Nevertheless, further studies are needed to compare deep neural networks in large datasets and automatic ROI detection of aneurysms (Page 7, line253-257; page 8, line 258-259). 

Reference

1.      Wei, Yu.; Kuiyuan, Yang.; Yalong, Bai.; Hongxun, Yao.; Yong, Rui. Visualizing and comparing AlexNet and VGG using deconvolutional layers. Proceedings of the 33rd international conference on machine learning.2016, New York, NY, USA. 

Reviewer 2 Report

The authors aimed to develop a computer-assisted detection system for small-sized aneurysm ruptures using a convolutional neural network based on images of three-dimensional digital subtraction angiography.

This study included retrospective data collection and a prospective study in patients harboring cerebral aneurysms, at multiple centers.

The study design is thorough, with inclusion-exclusion criteria, controls, statistical evaluation.

The topic is interesting, actual and involving the reader, it could be spread not only among the healthcare professionals but in a more wider audience.

Some points to be discussed:

1. The authors didn't mention any data on vasospasm after SAH - was it an obstacle? Did it interfere with the results somehow?

2. The DSA procedures - were they purely diagnostic, or interventional/therapeutic? What was the ratio/percentage of the procedures?

Because the percentage of diagnostic DSA decreases, depending upon the experience of a medical centre. Related to the matter the statement of the authors in Discussion, line 243 "Furthermore, DSA is also usually conducted.." - if you want to keep this statement, please support it with a reference.

Author Response

Comment 1: The authors didn't mention any data on vasospasm after SAH - was it an obstacle? Did it interfere with the results somehow?

Answer: Thank you for comments on our manuscript. In this study, we aimed to assess the rupture risk in small-sized aneurysms using 3D-images of initial diagnostic angiography. For SAH patients, DSA was usually performed within 24 hours after SAH. Vasospasm occurs most common from 3 to 10 days after SAH development. In addition, the diagnosis is usually done by routine daily transcranial Doppler or CT-angiography. Therefore, we did not mention vasospasm data in this study.

Comment 2: The DSA procedures - were they purely diagnostic, or interventional/therapeutic? What was the ratio/percentage of the procedures?

Answer: DSA procedures were all diagnostic to obtain detailed information on morphological characteristics of aneurysms.

Comment 3: Because the percentage of diagnostic DSA decreases, depending upon the experience of a medical centre. Related to the matter the statement of the authors in Discussion, line 243 "Furthermore, DSA is also usually conducted.." - if you want to keep this statement, please support it with a reference.

Answer: Thank you for your comments. We have removed these sentences from the manuscript.

Reviewer 3 Report

This is a well executed study and you should be pleased with the resulting data.  Please correct Line 50 where the "posterior circulating" instead of the "posterior cerebral" artery is referred to.

Author Response

Comment 1: This is a well executed study and you should be pleased with the resulting data.  Please correct Line 50 where the "posterior circulating" instead of the "posterior cerebral" artery is referred to.

 Answer: Thank you for your comments on our manuscript. Per your recommendation, we have corrected this to be ‘posterior cerebral artery’. 

Round  2

Reviewer 1 Report

The authors have addressed all my comments by adding more experiments and explanations in the revised manuscript (more deep learning models and HOG method). Therefore, I recommend accepting the manuscript without further amendments.